ecology

niche overlap, animal migration, scale, climate, weather

**Author for correspondence:**
Guillermo Fandos
e-mail: guillermo.fandos.guzman@hu-berlin.de

# Seasonal niche tracking of climate emerges at the population level in a migratory bird

Guillermo Fandos[1,2], Shay Rotics[3,4], Nir Sapir[5], Wolfgang Fiedler[6,7], Michael Kaatz[8], Martin Wikelski[6,7,9], Ran Nathan[3] and Damaris Zurell[1,2]

[1]Institute for Biochemistry and Biology, University of Potsdam, D-14469, Potsdam, Germany
[2]Geography Department, Humboldt-Universität zu Berlin, D-10099 Berlin, Germany
[3]Movement Ecology Lab, Department of Ecology, Evolution, and Behaviour, The Hebrew University of Jerusalem, Edmond J. Safra Campus at Givat Ram, 91904 Jerusalem, Israel
[4]Department of Zoology, University of Cambridge, Cambridge, UK
[5]Department Evolutionary and Environmental Biology and Institute of Evolution, University of Haifa, 3498838 Haifa, Israel
[6]Max Planck Institute of Animal Behavior, D-78315 Radolfzell, Germany
[7]Department of Biology, University of Konstanz, Konstanz, Germany
[8]Vogelschutzwarte Storchenhof Loburg e.V., Loburg, Germany
[9]Centre for the Advanced Study of Collective Behaviour, University of Konstanz, 78457 Konstanz, Germany

GF, 0000-0003-1579-9444; SR, 0000-0002-3858-1811; NS, 0000-0002-2477-0515;
WF, 0000-0003-1082-4161; MW, 0000-0002-9790-7025; RN, 0000-0002-5733-6715;
DZ, 0000-0002-4628-3558

Seasonal animal migration is a widespread phenomenon. At the species level, it has been shown that many migratory animal species track similar climatic conditions throughout the year. However, it remains unclear whether such a niche tracking pattern is a direct consequence of individual behaviour or emerges at the population or species level through behavioural variability. Here, we estimated seasonal niche overlap and seasonal niche tracking at the individual and population level of central European white storks (*Ciconia ciconia*). We quantified niche tracking for both weather and climate conditions to control for the different spatio-temporal scales over which ecological processes may operate. Our results indicate that niche tracking is a bottom-up process. Individuals mainly track weather conditions while climatic niche tracking mainly emerges at the population level. This result may be partially explained by a high degree of intra- and inter-individual variation in niche overlap between seasons. Understanding how migratory individuals, populations and species respond to seasonal environments is key for anticipating the impacts of global environmental changes.

## 1. Introduction

Each year, billions of individuals move over vast distances on a seasonal basis [1], including mammals, birds, fish and insects [2,3]. The urge to track suitable environmental conditions has been frequently proposed as one of the main mechanisms to explain these annual movements. In particular, long-distance migratory birds have high movement capacity and can travel between a wide range of possible environments, yet many species follow similar environmental conditions across the year [4]. However, such seasonal niche tracking is not a general rule, and many migrants do not track their environmental niche through seasons or even switch their niche between seasons [5–8]. It remains open why different species adopt different niche tracking strategies, and the mechanisms underlying the diverse relationships between movement and seasonal niche tracking are not well understood [9].

Here, we suggest that seasonal niche tracking may be best understood as a multi-scale phenomenon [10], and its study requires synthesis and integration

*Proc. R. Soc. B* **287**: 20201799

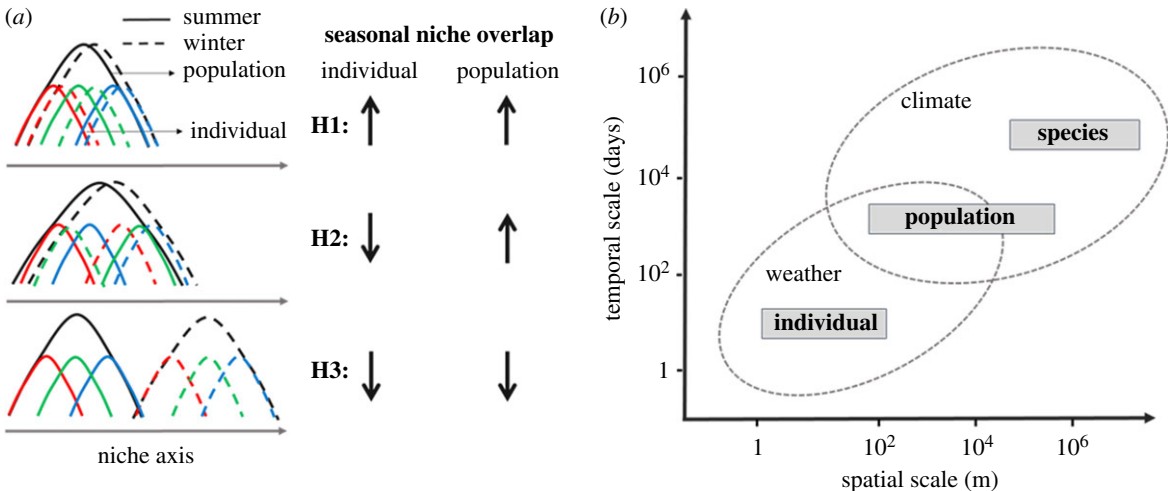

**Figure 1.** Conceptual overview of potential seasonal niche overlap patterns across ecological levels. (*a*) We present three hypotheses regarding how seasonal niche overlap of individuals (coloured lines) could scale up to seasonal niche overlap of populations (black lines). Upward and downward arrows indicate high and low niche overlap, respectively. H1: high niche overlap values for both ecological levels; H2: high niche overlap only at the population level; H3: low or no overlap at both population and individual levels. (*b*) Space-time diagram presenting the spatio-temporal scales at which ecological processes take place and how these are linked with environmental variability that takes place at multiple scales. (Online version in colour.)

of diverse ecological processes that occur at different spatial and temporal scales and across several hierarchical levels of ecological organization. Research in seasonal niche tracking tends to focus on the species level, analysing species-wide niche characteristics, and on comparisons between breeding site and wintering site conditions [4,5,11]. By contrast, seasonal niche tracking at the level of individuals and populations [12], and across the entire annual cycle has rarely been studied, largely because of limited data availability. With the progress in tracking technology, we are now obtaining unprecedented data on the year-round movement trajectories of animals, boosting the study of individual variability and flexibility of migratory movements [13]. So far, only a few studies have investigated the link between seasonal environmental dynamics and movement strategies, mostly focusing on resource selection [14,15]. These works indicate that individual resource selection can drive population-level movement patterns [16]. Yet, it has not been tested how such behaviours scale up to seasonal climatic niche tracking. As migratory animals are declining globally [17], there is an urgent need to understand how climate affects seasonal movements at multiple ecological levels.

We propose three alternative hypotheses of how seasonal niche tracking could be linked across ecological levels (figure 1*a*). First, niche tracking may be an individual-level property that scales up to the population and species level (H1). Second, even if individuals do not track seasonal climates, niche tracking may emerge at higher ecological levels through variability in individual seasonal niches, including within-individual behavioural variation and between-individual interactions (H2) [18–21], as observed in herbivores tracking seasonal foliage dynamics [19]. Lastly, niche tracking may be absent at any ecological level (H3; figure 1*a*). As a complicating factor, the response to environmental factors may vary across ecological levels (figure 1*b*). At the species level, we would expect climate or vegetation gradients to be more important [4], while at the individual level, migratory animals may rather respond to local scale cues such as weather, or a combination of local- and broad-scale factors [13]. Decoupling individual and population-level niche

tracking behaviours would provide valuable insights how migrants select suitable environments throughout the annual cycle and will improve our predictions of how migratory individuals and populations will respond to environmental changes at multiple scales, including global climate warming.

In this study, we aim to understand seasonal niche tracking of migratory birds at multiple ecological scales. We focus on white storks (*Ciconia ciconia*) breeding in central Europe, which are soaring, obligatory social long-distance migrants. White storks mostly breed in Eurasia and migrate to central and southern Africa along two main flyways, east and west of the Mediterranean Sea. Movement data were available from 35 eastern- and western-migration individuals in the years 2012–2017, which were tagged in their breeding grounds in central Germany (figure 2). All individuals were adults, to avoid including data related to the poor flight skills of juvenile storks [22]. We analysed eastern and western migrants as different populations because migration patterns (distances, seasonal ranges) and breeding phenology (spring arrival and breeding starting date) vary greatly between both migratory flyways with potentially significant consequences for seasonal niche dynamics [22,23].

Our main objective is to test the emergence of seasonal niche overlap patterns at multiple ecological levels (figure 1*a*). Specifically, we (i) quantify and compare individual and population-level niche overlap across four seasons (spring, summer, autumn, winter) using ordination techniques that account for spatial autocorrelation in environmental conditions between regions. Based on similarity tests, we (ii) quantify and compare individual and population-level niche tracking across four seasons by comparing the estimated niche overlap against null models. Niche tracking occurs if the observed niche overlap is higher than expected by chance, given the available environmental conditions. Finally, (iii) we assess variation in niche overlap and niche tracking estimates between individuals and within individuals as well as between populations (i.e. between eastern and western migrants) and within populations. Knowledge of the presence or absence of such variation is important to understand the potential mechanisms behind the seasonal niche

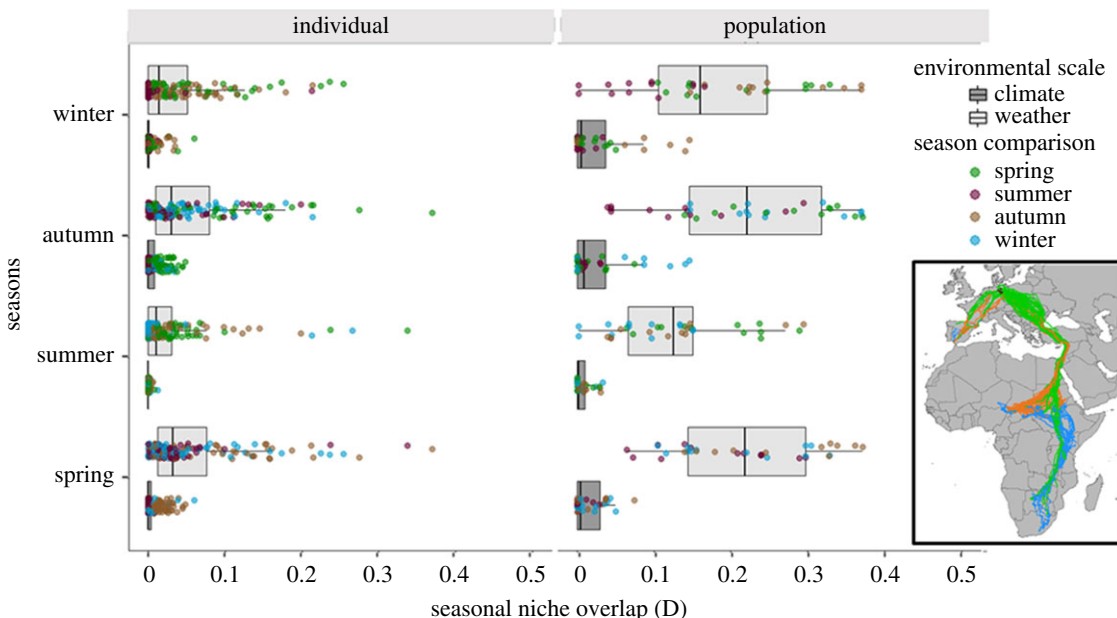

**Figure 2.** Seasonal niche overlap. Observed niche overlap (Schoener's *D*) between seasons at the individual versus population level and with regard to weather versus climate data. The colour of the boxplot represents the environmental dataset (weather or climate), and the points show the respective overlap value for each season (dark purple: summer; orange: autumn; blue: winter; green: spring). The overlap metrics *D* were estimated in three-dimensional environmental space with the axes representing temperature, precipitation and NDVI. The inset shows the seasonal movement tracks of 35 individual white storks during 2012–2017. Coloured lines indicate the four seasons (dark purple: summer; orange: autumn; blue: winter; green: spring). (Online version in colour.)

tracking patterns [21]. Additionally, to account for the fact that ecological and environmental processes are scale-dependent - (figure 1*b*), we estimate seasonal niche overlap and tracking both in terms of climate (long-term average over greater than 15 years) and weather data (defined as the fine-scale conditions over a short period; less than 20 days).

## 2. Material and methods

### (a) Tracking data
We trapped 62 adult white storks in the state of Saxony-Anhalt, Germany, and equipped them with solar GPS-ACC transmitters (e-obs GmbH; Munich, Germany) that weighed 55 g including harness, which is roughly 2% of the average stork's weight (see [24]). The transmitters recorded GPS fixes every 5 min when solar conditions were good (95% of the time) or every 20 min otherwise. For subsequent analyses, we randomly selected a set of 100 GPS locations per day to estimate the seasonal niche and to avoid over-fitting the data to some locations. This procedure reduces potential biases through spatial and temporal autocorrelation in the point records and behaviours (e.g. breeding, foraging, resting) [25,26]. As we are interested in the niche overlap across seasons, we excluded all individuals that were tracked for less than one entire year (used data range: 1–5 years). In total, we used 35 white stork individuals that provided 5 704 483 georeferenced locations (range per individual 39 741–327 642). To estimate the environmental niche of each season, we divided all selected points into sets of two months along the year: spring (March–April), summer (June–July), autumn (September–October) and winter (December–January). The one-month gap between seasons controls for phenological variation (e.g. in breeding and migrating) between individuals and populations (figure 1).

### (b) Environmental data
To account for scale dependence in ecological processes, we separately analysed niche overlap and niche tracking for the weather and climate, and thus derived two different environmental datasets.

For weather data, all selected points were annotated with environmental data of temperature (Land Surface Temperature & Emissivity 1-km Daily Terra; MOD11A1 V6), precipitation (ECMWF Interim Full Daily SFC-FC Total Precipitation; 0.75 deg.; 3 hourly) and Normalized Difference Vegetation Index (NDVI; MODIS Land Vegetation Indices 1 km 16 days Terra) using the Env-DATA track annotation tool of MoveBank [27]. For the climate data, we used long-term averaged monthly temperature and precipitation patterns for the time period 1979–2013 at 1 km resolution (CHELSA [28]), and monthly NDVI for the time period 1982–2000 (GIMMS AVHHR Global NDVI [29]), and extracted the values of each variable for all selected points using the 'raster' package [30].

### (c) Seasonal niche overlap
In the seasonal niche overlap analyses, we considered three environmental niche axes: temperature and precipitation representing climatic niche axes, and NDVI representing resource use. We calculated environmental niche overlap between seasons by means of ordination [31]. This approach estimates the density of GPS locations along the environmental niche axes and corrects this for the density of the total available environment (the background) along the same niche axes (by dividing the two densities). Niche overlap between pairs of two seasons is then quantified using Schoener's *D* metric [32], which varies between 0 (no overlap) and 1 (perfect overlap). Here, we quantified niche overlap between pairs of seasons at the individual and at the population level, and separately for weather and climatic data.

In both cases, with weather and climate, we considered four different environmental sets to analyse niche overlap: (i) all environmental variables, including temperature + precipitation + NDVI, (ii) temperature, (iii) precipitation and (iv) NDVI; this allowed us to examine the contribution of each variable to patterns of seasonal niche overlap.

We spatially thinned the GPS locations to avoid spatial autocorrelation by randomly removing locations that were within the same 1 km cell [31]. Background data were derived separately for eastern and for western migrants. For this, we first calculated the 100% minimum convex polygon (MCP) containing all eastern-migrant or all western-migrant GPS locations. Second, we

placed a buffer of 300 km around these MCPs, which corresponds to the maximum white storks home range size [33] observed during staging and wintering. We sampled 10 000 background points from these areas to consider all the environmental conditions available for each migratory flyway throughout the year. Finally, we annotated these background locations with climate and weather data. For climate, we extracted the environmental information (temperature, precipitation and NDVI) in each season for all background points. For weather, the background locations were assigned a season-specific random date in the same temporal range as the recorded GPS locations and then were annotated with environmental data of temperature, precipitation and NDVI using the Env-DATA track annotation tool of MoveBank [27].

To estimate the density of GPS locations and density of background locations in the three-dimensional environmental space, we calculated a multivariate kernel density smoother to allow consideration of the full environmental variation along all three niche axes (for more details, see [4,31]). At the individual level, the occurrence (GPS) density was calculated separately for each individual and year. At the population level, occurrence density was calculated separately for each migratory flyway and year. Additionally, we calculated the univariate densities along the single environmental variables to examine the individual contribution of each variable to patterns of seasonal niche overlap. Before quantifying niche overlap, all occurrence (GPS) densities were divided by the background densities to correct for differences in the relative availability of environments. We then calculated Schoener's $D$ metric for all pairs of seasons, at the individual and at the population level, for weather and climate data, and for the three-dimensional environmental niche space as well as the three separate univariate niche axes. We used unequal variance $t$-test (Welch test [34]) to explore the difference between the niche overlap estimates for climate and weather.

## (d) Seasonal niche tracking

We applied similarity tests using a null model approach to quantify seasonal niche tracking at the individual level and the population level. To this end, the observed niche overlap estimates were compared against simulated niche overlap estimates obtained from randomized data. By this, we were able to assess whether niche overlap between pairs of seasons was higher than expected by the background environmental conditions. Specifically, in similarity tests, the occurrence points of one season, out of a pair of two seasons, are permuted randomly in the three-dimensional environmental space, and the simulated densities are divided by the background densities. Then, the overlap between the resulting simulated niche is compared to the observed niche of the other season. We repeated this procedure 200 times per pair of seasons (100 permutations where the niche of one season was assumed to be the observed niche while the niche of the other season was permutated, and another 100 permutations vice versa [4]). For each permutation, we calculated the simulated niche overlap between simulated and observed niche using the Schoener's $D$ metric. An individual or a population was defined to track its niche between a pair of seasons if the observed value of the Schoener's $D$ metric was greater than 95% of the simulated values, meaning that the individual/population inhabits more similar environmental conditions across the two seasons than expected by chance given the available environment [31]. All analyses were performed in the R platform [35] using the package 'Ecospat' [36] and the scripts provided by Broennimann et al. [31] and Zurell et al. [4].

## (e) Variation in seasonal niche overlap and tracking

Last, we aimed to determine the variation in seasonal niche overlap and seasonal niche tracking within and between individuals and populations. For this, we used mixed-effects models to partition the variation into within and between-individual components as well as within and between-population components [37].

We fitted four models to partition the niche overlap variation for each ecological level (population versus individual) and each environmental scale (climate versus weather), using the standardized niche overlap as a response variable and assuming Gaussian error distributions. Analogously, we fitted four models to partition the niche tracking variation, using the categorical variable of niche tracking 'yes/no' as a response variable and assuming a categorical error distribution. For the individual variation, we used the nested effect of the individual within the year as random effects, and to statistically control for potential sources of variation, we included the migratory flyway as a fixed effect. This approach permits us to estimate the between-individual (individual mean) and within-individual (deviation from the individual mean) variation of the environmental niche overlap and niche tracking. For the population variation, we used the nested effect of the population (eastern or western) within the year as random effects and only the intercept as a fixed effect. Here, we estimate the between-population (population mean) and within-population (deviation from the population mean) variation of the environmental niche overlap and niche tracking.

From all these models, we calculated the repeatability of the niche overlap and niche tracking, respectively. Repeatability represents the proportion of variance that is attributable to differences between individuals (or populations) and provides a standardized estimate that can be compared across all models [37]. For this, we first extracted the between-individual (or between-population) variance $V_B$ from the models defined as the variance across the random intercepts of the individuals (or populations) and the within-individual (or within-population) variance $V_W$ defined as the 'residual error' of the model. Repeatability is given by $V_B/(V_B + V_W)$ [37] and scales from 0 to 1, with 0 indicating that all the variance is within individuals/populations, and 1 indicating that all the variance is between individuals/populations [38]. Confidence intervals around repeatability values were calculated following [38].

All models were fitted in R v. 4.0.0 [35] with the package 'MCMCglmm' [39] and had a relatively uninformative expanded prior for the variance structures (R and G) with $V = 1$ (prior expectation) and nu = 0.002 (belief parameter) for the R-structure, and with $V = 1$, nu = 0.02, alpha.mu = (0, 0) and alpha.$V = 1000$ for the G-structure. Models were run for 500 000 iterations with a thinning rate of 100 and a burn-in of 50 000. We estimated 95% credible intervals from the posterior distribution, and any interval not including 0 was considered significant. We checked trace plots for adequate mixing and all autocorrelations were less than 0.1.

## 3. Results

## (a) Seasonal niche overlap

We found low individual-level niche overlap for both climate (mean = 0.004, range 0.00–0.06) and weather (mean 0.04, range 0.00–0.36; figure 2; electronic supplementary material, figure S1). At the population level, we found a similar pattern of comparably low seasonal niche overlap for climate (mean 0.019, range 0.000–0.146), but larger overlap for weather (mean 0.186, range 0.00–0.37; figure 2; electronic supplementary material, figure S1). Overall, we estimated greater niche overlap for weather than for climate ($t$-test; population level: $t = -19.086$, d.f. = 172.57, $p$-value < 0.001; individual level: $t = -16.606$, d.f. = 645.22, $p$-value < 0.001), and greater niche overlap at the population level than at the individual level ($t$-test; climate scale: $t = -5.7618$, d.f. = 144.59, $p$-value < 0.001;

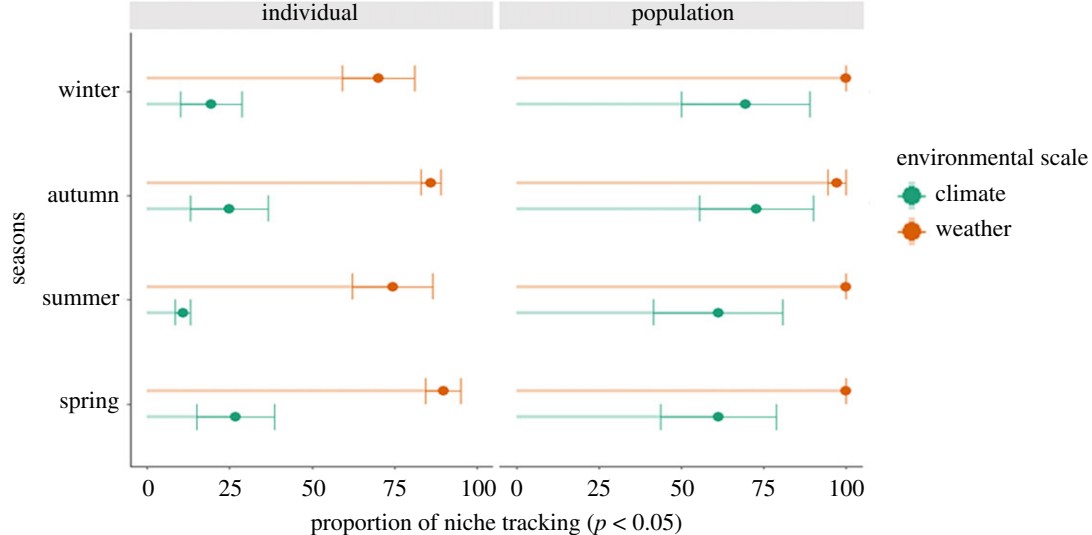

**Figure 3.** Seasonal niche tracking. The proportion of significant niche tracking across seasons at individual and population level and for weather and climate, respectively. Niche tracking was analysed using similarity tests (with $n = 200$ randomizations and a significance level of $\alpha = 0.05$). Colours indicate the environmental dataset used (weather/climate). The niche tracking proportion was estimated in three-dimensional environmental space with the axes representing temperature, precipitation and NDVI. (Online version in colour.)

weather scale: $t = -16.475$, d.f. = 167.19, $p$-value < 0.001). When comparing niche overlap separately for the different environmental variables, climatic niche overlap was highest for NDVI, and weather niche overlap was highest for precipitation (electronic supplementary material, table S1).

### (b) Seasonal niche tracking

In our niche similarity tests with 200 randomizations, we found high proportions of population-level niche tracking for both climate and weather variables (66.11% of populations were tracking climate, and 99.30% were tracking weather; figure 3). By contrast, at the individual level, we found much lower niche tracking for climate (20.46%; figure 3), but high niche tracking for weather (99.30%; figure 3). When separately analysing the different weather and climate variables of temperature, precipitation and NDVI, niche tracking was generally lower compared to the multi-dimensional niche analyses (electronic supplementary material, table S2).

### (c) Variation in seasonal niche overlap and niche tracking

The partition of the variance between and within individuals show similar patterns for the niche overlap and niche tracking behaviour (figure 4; electronic supplementary material, table S3). In both cases, we found low repeatability with the weather and climate variables (repeatability <0.11; electronic supplementary material, table S3) indicating that the variations in seasonal niche overlap and in seasonal niche tracking are mainly attributable to within-individual variability. Differences among years had a minor effect on individual-level niche overlap and niche tracking, respectively. Interestingly, the effect of the migratory flyway varied across the considered cases. Specifically, western migrants showed less climatic niche overlap than their eastern counterparts and no differences in weather niche overlap. By contrast, we found that western migrants were tracking weather significantly better than eastern migrants while differences in climate niche tracking were insignificant (figure 4).

At the population level, we found similar effects of within and between-population variation on niche overlap and niche tracking for both climate and weather (figure 4; electronic supplementary material, table S3). For both niche overlap and niche tracking, we found high repeatability (repeatability greater than 0.95; electronic supplementary material, table S3). Overall, between-population variation was higher for the climate than for weather (figure 4; electronic supplementary material, table S3). Specifically, eastern migrants showed much higher climate niche tracking compared to western migrants (electronic supplementary material, figure S2). Differences among years had a minor effect on the population-level niche overlap and niche tracking.

## 4. Discussion

Our study contributes to a better understanding of how migratory birds may shift or conserve their niches throughout the year. By explicitly analysing niche overlap and niche tracking at multiple ecological and spatio-temporal scales, we were able to link individual environmental use with population-level patterns. Results indicated that seasonal niche tracking of weather and climate vary across ecological levels. Thus, seasonal niche tracking is a complex phenomenon, with different driving mechanisms operating across spatial and temporal scales.

We found that climatic niche tracking in white storks emerges only at the population level. Thus, our results emphasize that different ecological levels have different spatial and temporal scales of response [40,41]. Such emergent population-level patterns have also been reported in other contexts, for example, showing an effect of individual-level decisions on survival and reproductive productivity [38,39]. In our analyses, individuals only tracked weather conditions, probably because white storks as typical soaring migrants actively follow fine-scale environmental conditions to reduce movement costs [22]. By contrast, populations tracked both climate and weather (figure 3). Both individual-level and population-level seasonal niche overlap were low compared to previous species-level studies [4]. This result may come from seasonally

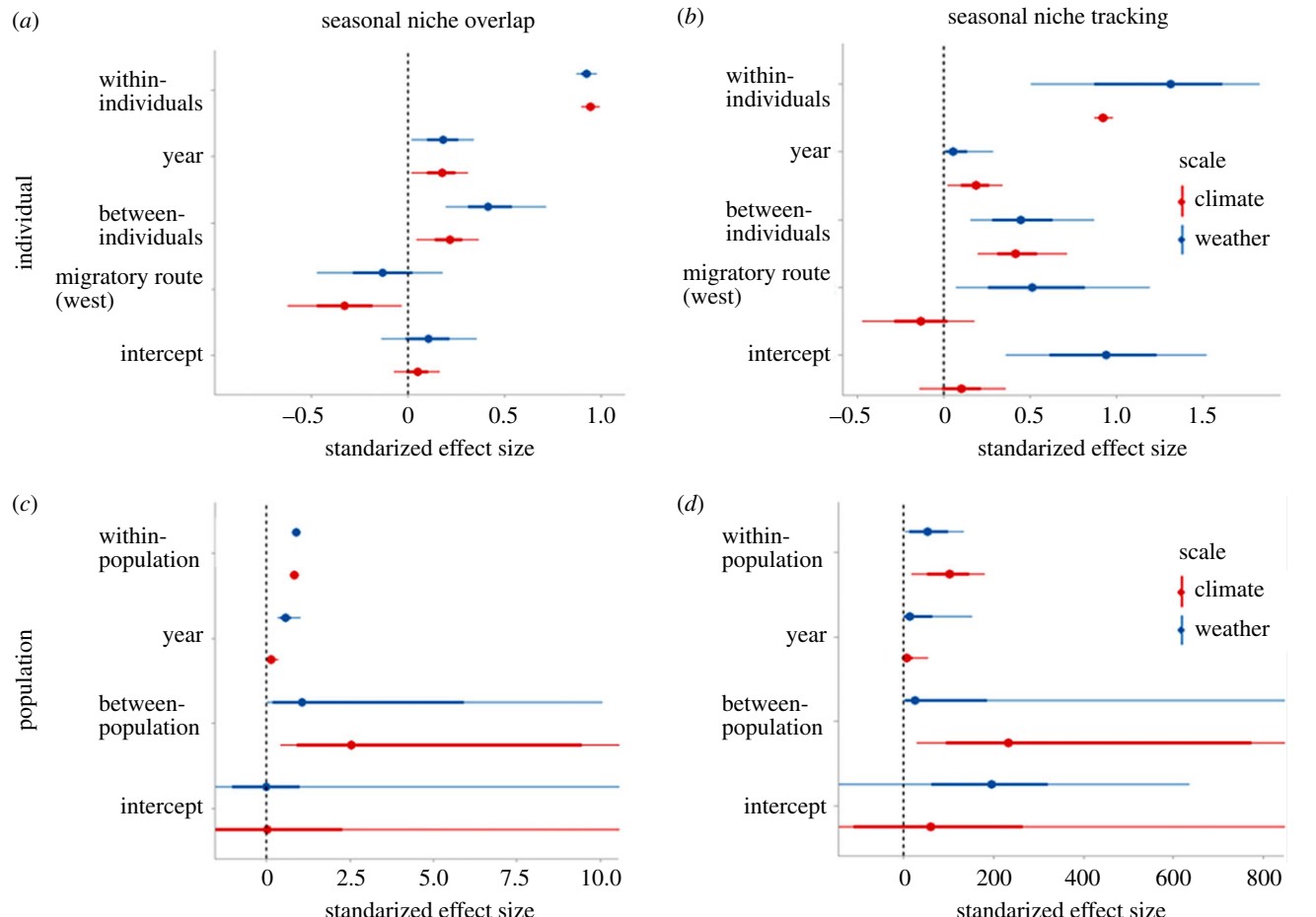

**Figure 4.** Variation in seasonal niche overlap and niche tracking variability estimation. Standardized coefficients from MCMC generalized linear mixed models of the niche overlap (*a,c*) and seasonal niche tracking (*b,d*) at the individual level (*a,b*) and population level (*c,d*) for weather and climate, respectively. Points indicate the mean coefficients, thick bars indicate the posterior standard deviations and the thin lines the 95% credible intervals of the coefficient values. Red and blue line colours indicate climate and weather data, respectively. In (*b*), we divided the standardized effect size by 100 for weather data for a better presentation of the results. The niche overlap and niche tracking were estimated in three-dimensional environmental space with the axes representing temperature, precipitation and NDVI. The reference level for the categorical variable migratory flyway is 'eastern migrants'. The overlap of the error bars with the dashed line at zero indicates that the effect of the parameter is not statistically significant (i.e. $p > 0.05$). (Online version in colour.)

complex environment relationships with opportunistic foraging behaviour within and between seasons [41,42]. Nevertheless, high proportions of individuals and the population tracked their environmental niche throughout the year more than expected by chance. By contrast, seasonal niche tracking along single environmental variables was lower (0.34–28.91%; electronic supplementary material, table S2). Thus, seasonal niche tracking of both, weather and climate, is a complex combination of multiple environmental dimensions, including local food resource availability (NDVI) and ecophysiological drivers (precipitation and temperature).

Our results further showed that the western migrants that winter in the Iberian Peninsula (figure 2) exhibit much lower climate niche tracking than the eastern migrants resulting in high between-population variation (electronic supplementary material, figure S2 and table S3; figure 4). This loss of climate niche tracking behaviour in the western migrants might be a consequence of the increasing number of resident individuals in the Iberian Peninsula due to recent anthropogenic changes and the constant spatio-temporal availability of food resources (e.g. landfills) in this region [22,43]. Additionally, we found high within-individual compared to between-individual variation repeatability in seasonal niche overlap and niche tracking for both climate and weather and independent of the migratory

flyway (figure 4). The high individual plasticity might be the source of emergent patterns at the population level, which can be the signature of adaptive mechanisms [21]. Individual variation might result from fluctuations in the internal state of individuals (energy level [44], health status [45], experience [46]) or extrinsic factors experienced by them [43] (e.g. food availability, day length, meteorological conditions). Gilroy *et al.* [47] have discussed in detail how plasticity in migratory movements and migratory strategy can affect the ecological and evolutionary success of populations and species in the face of environmental change. Empirically, the relationship between migratory patterns and vulnerability to environmental change has typically been explored at the population level, neglecting variation among and within individuals. However, information about the plasticity of seasonal niche overlap and niche tracking at different hierarchical levels (individuals, populations, species) can provide better insights about the resilience of species and populations to environmental change [48,49]. The ability of individuals to directly, and adaptively, adjust their level of niche overlap or niche tracking to prevailing environmental conditions, could trigger behavioural responses to global change on an ecological rather than evolutionary time-scale.

As a small limitation, we analyse niche tracking at the population level with only two populations (albeit during

several years), excluding the species level from our analysis. It would be desirable in the future to incorporate more populations from the entire breeding range of the species and assess niche overlap and niche tracking across all ecological levels, from individual to the species level (figure 1*b*). Also, future analyses should be extended to more species and geographic regions to ascertain whether the consequences of individual variation are generalizable to different migratory species and different movement behaviours (e.g. non-soaring birds) and across ecosystems.

The use of tracking data from multiple years, in tandem with a multi-scale approach, has the potential to disentangle the different hierarchical drivers of migration. Improving our understanding of diverse aspects of the niche across the annual cycle is essential to forecast how migratory individuals, populations and species will respond to changing environmental conditions. The result that individual white storks track weather but not climate, together with the documented increase in European overwintering storks and the shortening of migratory distances [49], highlights the ability of the storks to adapt to rapid human-induced environmental changes. Thus, niche tracking seems to be a bottom-up mechanism, at least in long-lived and opportunistic migrants such as white storks. Overall, the potential individual flexibility of using different environments along the year, and its costs and benefits, may have important implications for fast microevolutionary changes in migratory patterns that are fundamental to successfully cope with global change.

Ethics. This research was carried out with approvals from (a) the National Administrative Office of Sachsen-Anhalt, Germany, Division of Nature Conservation, 407.3.3/255.13-2248/2, (b) the State Office for Environment, Health and Consumer Protection of Brandenburg, Germany, V3-2347-8-2012.

Data accessibility. The 100 GPS-ACC data locations randomly selected for each individual and day, together with their respective climate and weather variables are available from the Dryad Digital Repository: https://doi.org/10.5061/dryad.x0k6djhgx [50].

Authors' contributions. G.F. and D.Z. designed the study. S.R., N.S., W.F., M.K., M.W. and R.N. collected the data and contributed to data preparation. G.F. ran the analyses and drafted the manuscript. All authors contributed significantly to the revision of the manuscript.

Competing interests. We declare we have no competing interests.

Funding. G.F. and D.Z. were supported by Deutsche Forschungsgemeinschaft (DFG) under grant agreement No. ZU 361/1-1. We further acknowledge the generous funding of DIP grant no. (DFG) NA 846/1-1 and WI 3576/1-1 to R.N. and M.W.

Acknowledgements. We would like to thank Florian Jeltsch, Andrea Flack, for their help with the tracking data.

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
