## [Reviewer comments · Proceedings of the Royal Society B: Biological Sciences]

Review History

RSPB-2020-0658.R0 (Original submission)

Review form: Reviewer 1 (Camila Gómez)

Recommendation

Major revision is needed (please make suggestions in comments)

Scientific importance: Is the manuscript an original and important contribution to its field?

Good

General interest: Is the paper of sufficient general interest?

Good

Quality of the paper: Is the overall quality of the paper suitable?

Acceptable

Is the length of the paper justified?

Yes

Should the paper be seen by a specialist statistical reviewer?

No

Do you have any concerns about statistical analyses in this paper? If so, please specify them explicitly in your report.

No

It is a condition of publication that authors make their supporting data, code and materials available - either as supplementary material or hosted in an external repository. Please rate, if applicable, the supporting data on the following criteria.

Is it accessible?

Yes

Is it clear?

Yes

Is it adequate?

Yes

Do you have any ethical concerns with this paper?

No

Comments to the Author

This manuscript describes how niche tracking varies across spatio-temporal scales and between individuals and populations of White Storks. The detailed multi-year tracking dataset that they used is very impressive and allows for unprecedented detail to be included in these types of analysis. I thought the paper was very interesting and has great potential to be a valuable contribution in its field.

Given this amazing dataset, I was a bit disappointed by the main figures on the ms, and think that the authors should explore other options to improve the visualization their results. In my opinion, improving those figures will make the paper much more compelling and its main message clearer to a wider readership. I think it is critical that authors show the variability in their dataset explicitly, hence not just graphing means, but also raw data points, and provide a clearer framework for the interpretation of their figures.

I have provided some details below and hope they can serve to improve the manuscript.

Notes:

Title – I would change ‘migratory birds’ to ‘a migratory bird’. This is a single species study.

Figure 2 (and figure SM4). Very cool but completely unnecessary in the context of your manuscript. They do not add anything to your message. Maybe a way to include a map of tracks could be if you color code the tracks by season. That map could then be included as a small side panel within a main figure of niche overlap, just showing where geographically are the storks during each period in their annual cycle. You could add a small illustration of a White Stork to make it even better!

Figs 3 and 4. These are the most important figures but I had a lot of trouble interpreting them. I like the fact that they are circular, but I don’t like that they don’t make the variability of the data explicit, and frankly, I did not understand the idea with the line colors and how they connect to each other. I think that authors could keep these (maybe as a side or complementary panel to the figure I am suggesting), and providing a clearer explanation of their interpretation. I strongly suggest authors explore other attractive ways to visualize their data. For instance, they could explore creating a kernel density circular plot for individuals and populations, showing all the individual data points and estimating the core D scores from the seasonal distributions. Points would be color and spatially coded on the circles by season of the year (date would determine the cartesian angle of each point) and D value would determine the distance from the center of the circle. I imagine four ‘bubbles’, one for each season, and with points concentrating around the mean D score for the season, but with the variability explicitly shown by all the data points.

Showing a spatial representation of this variability is a much more transparent and thorough way to present these data, and I think will be better for interpretation.

Fig 5. Please overlay all the jittered data points on the boxplots.

Line 260 – ‘tracking its climatic niche’ should be ‘tracking their climatic niche’

Line 272 – 284. The point raised in this paragraph should be emphasized more as it is one of the most novel aspects of this study. There has rarely been data available to look and intra individual and inter individual variation in migration strategies, so they should make a strong point about that here.

Line 299 – Individuals should be ‘individual’

Review form: Reviewer 2

Recommendation

Major revision is needed (please make suggestions in comments)

Scientific importance: Is the manuscript an original and important contribution to its field?

Acceptable

General interest: Is the paper of sufficient general interest?

Acceptable

Quality of the paper: Is the overall quality of the paper suitable?

Marginal

Is the length of the paper justified?

Yes

Should the paper be seen by a specialist statistical reviewer?

No

Do you have any concerns about statistical analyses in this paper? If so, please specify them explicitly in your report.

Yes

It is a condition of publication that authors make their supporting data, code and materials available - either as supplementary material or hosted in an external repository. Please rate, if applicable, the supporting data on the following criteria.

Is it accessible?

Yes

Is it clear?

Yes

Is it adequate?

Yes

Do you have any ethical concerns with this paper?

No

Comments to the Author

This paper uses detailed tracking data to assess the variability of White Stork migration and whether individuals, as well as the population as a whole, tracks climatic conditions throughout the annual cycle. The objective of the paper is valuable and the potential for interesting insights is strong. Overall, the presentation of the results and the writing was pretty good. However, I felt

that the paper lacked important detail and explanation in several key areas that make it very difficult for the reader to assess the conclusions and the impact of this work. My comments are intended to point out areas that need greater explanation or thought.

Major comments:

First, the authors assess the impact of weather and climate, but they don't adequately explain the biological difference between the two or how their data allow them to make this distinction for the birds.

Second, much of the paper is focused on contrasting niche overlap with niche tracking. However, the distinction between these two concepts is not explained. Even as someone who is generally familiar with niche modelling and has read most of the papers on niche tracking in migratory birds, I had difficulty following the distinctions the authors were making between these two metrics (niche overlap and niche tracking). Consequently, it is difficult to assess whether the difference between Figure 3 and Figure 4 is actually a biological difference, or simply a consequence of different methodologies that are expected to produce somewhat different results. This lack of clarity is due to the fact that the authors do not do an adequate job of connecting their methodology with their hypotheses. More explanation is needed. I did like the creative design of Figures 3 and 4, I just found the insights they were supposed to provide to be elusive.

Third, much of the results are focused on individual- versus population-level differences. However, as far as I can tell, the authors never explain what kind of individual variation they are measuring. I fail to understand whether the individual variation is across years, or just between seasons but within a year. And, if a bird was tracked for multiple years but the focus is on between-seasons, how was cross-year data treated in terms of non-independence?

Fourth, given that this paper focuses on one species, more biology of this species is needed in the paper. For example, Figure 2 shows some interesting dynamics going on in central Africa, which seem like they could have implications for tracking of climatic conditions, but this is never explained. Do White Storks have genetically encoded migration as juveniles, or do they follow their parents?

Finally, this comment represents a broader issue for the field that I think the authors should consider. The authors write on line 282-284: "The large within and between- individual variation found here indicate that population-level analyses could hide some flexibility at the individual level that could play an essential role in determining the resilience of species and populations to environmental change [48,49]". OK...but it seems to me that every study that tracks individual migratory birds, which usually have sample sizes of 50 or less, identify variation in individual behavior and come to the conclusion that migration is more variable than typically assumed. But, variable compared to what? In a small sample, there will obviously be variation in how birds get to Africa and back. But the question is, what does the distribution of this behavior for the species as a whole look like? So if the point of this paper is to provide insight into this important question by identifying patterns of scale, I think it is not adequate to simply conclude that population-level analyses hide individual variation. In other words, I'm not clear on how this study accomplishes its important goals of using individual variation to reveal emergent patterns. Perhaps it's just a matter of a better explanation.

Minor comments:

I was confused about how the two wintering populations were treated. In the introduction it suggests that these were lumped into one analysis, but in the results and Discussion (e.g., line 286) the authors start talking about the differences between the populations again as though this dimension is an advantage of the study. This was unclear.

Line 141-144. I don't understand how the background space was created around the tracks. What is the space represented by "all the spatial and temporal extent that GPS locations included in this study"? A continent? Or less? Given the big debates in the niche modeling literature about selection of background points, this seems pretty important to explain.

Line 150-161: I just found this paragraph to be difficult to parse and the writing not very clear. It would be better if the authors would connect each method back to the hypotheses they are testing, which are never really stated.

Figure 1a: I found this confusing, because the middle panel's illustration looks like low population overlap but the arrows label it as high population overlap. So I was never really clear on how the study identifies patterns of scale.

Decision letter (RSPB-2020-0658.R0)

27-Apr-2020

Dear Dr Fandos:

I am writing to inform you that your manuscript RSPB-2020-0658 entitled "Seasonal niche tracking of climate emerges at the population level in migratory birds" has, in its current form, been rejected for publication in Proceedings B.

This action has been taken on the advice of referees, who have recommended that substantial revisions are necessary. With this in mind we would be happy to consider a resubmission, provided the comments of the referees are fully addressed. However please note that this is not a provisional acceptance.

Sincerely,
Dr Locke Rowe
<mailto:proceedingsb@royalsociety.org>

Associate Editor

Comments to Author:

This is an interesting study seeking to examine whether patterns of climatic niche tracking across seasons often seen in migratory animals reflect tracking of conditions by individuals or rather reflect an emergent property of populations within which individuals may show varying strategies. Solving this question would appear to be of broad relevance to understanding issues in the ecology, evolution, and conservation of migrants. Despite the appealing overall question (and the impressive tracking data set for White Storks), and that reviewers found considerable merit in the study, the manuscript as presented falls short in terms of conveying a clear message for various reasons. As noted by the reviewers, figures are unclear or seemingly not directly relevant to making the main points; they might be, but several improvements are necessary to make them understandable and compelling. The reviewers focused on figures showing results, but I also felt that the conceptual figure 1 could be substantially improved not only in terms of graphics but with a more informative legend so that alternative scenarios could be readily understood. In addition, as noted by one of the reviewers, important clarifications are required so that the reader can properly assess the conclusions and the impact of this work. Specific points include, but are not limited to, the distinctions between weather and climate and niche overlap and niche tracking, and clarity about the kind of individual variability being measured. To the fine comments provided by reviewers, I would add that framing the introduction (and the rest of the paper) around clearly articulated hypotheses with their specific predictions would likely help considerably to present a more coherent story.

Reviewer(s)' Comments to Author:

Referee: 1

Comments to the Author(s)

This manuscript describes how niche tracking varies across spatio-temporal scales and between individuals and populations of White Storks. The detailed multi-year tracking dataset that they used is very impressive and allows for unprecedented detail to be included in these types of analysis. I thought the paper was very interesting and has great potential to be a valuable contribution in its field.

Given this amazing dataset, I was a bit disappointed by the main figures on the ms, and think that the authors should explore other options to improve the visualization their results. In my opinion, improving those figures will make the paper much more compelling and its main message clearer to a wider readership. I think it is critical that authors show the variability in their dataset explicitly, hence not just graphing means, but also raw data points, and provide a clearer framework for the interpretation of their figures.

I have provided some details below and hope they can serve to improve the manuscript.

Notes:

Title - I would change 'migratory birds' to 'a migratory bird'. This is a single species study.

Figure 2 (and figure SM4). Very cool but completely unnecessary in the context of your manuscript. They do not add anything to your message. Maybe a way to include a map of tracks could be if you color code the tracks by season. That map could then be included as a small side panel within a main figure of niche overlap, just showing where geographically are the storks during each period in their annual cycle. You could add a small illustration of a White Stork to make it even better!

Figs 3 and 4. These are the most important figures but I had a lot of trouble interpreting them. I like the fact that they are circular, but I don't like that they don't make the variability of the data explicit, and frankly, I did not understand the idea with the line colors and how they connect to each other. I think that authors could keep these (maybe as a side or complementary panel to the figure I am suggesting), and providing a clearer explanation of their interpretation. I strongly suggest authors explore other attractive ways to visualize their data. For instance, they could explore creating a kernel density circular plot for individuals and populations, showing all the individual data points and estimating the core D scores from the seasonal distributions. Points

would be color and spatially coded on the circles by season of the year (date would determine the cartesian angle of each point) and D value would determine the distance from the center of the circle. I imagine four 'bubbles', one for each season, and with points concentrating around the mean D score for the season, but with the variability explicitly shown by all the data points. Showing a spatial representation of this variability is a much more transparent and thorough way to present these data, and I think will be better for interpretation.

Fig 5. Please overlay all the jittered data points on the boxplots.

Line 260 - 'tracking its climatic niche' should be 'tracking their climatic niche'

Line 272 - 284. The point raised in this paragraph should be emphasized more as it is one of the most novel aspects of this study. There has rarely been data available to look and intra individual and inter individual variation in migration strategies, so they should make a strong point about that here.

Line 299 - Individuals should be 'individual'

Referee: 2

Comments to the Author(s)

This paper uses detailed tracking data to assess the variability of White Stork migration and whether individuals, as well as the population as a whole, tracks climatic conditions throughout the annual cycle. The objective of the paper is valuable and the potential for interesting insights is strong. Overall, the presentation of the results and the writing was pretty good. However, I felt that the paper lacked important detail and explanation in several key areas that make it very difficult for the reader to assess the conclusions and the impact of this work. My comments are intended to point out areas that need greater explanation or thought.

Major comments:

First, the authors assess the impact of weather and climate, but they don't adequately explain the biological difference between the two or how their data allow them to make this distinction for the birds.

Second, much of the paper is focused on contrasting niche overlap with niche tracking. However, the distinction between these two concepts is not explained. Even as someone who is generally familiar with niche modelling and has read most of the papers on niche tracking in migratory birds, I had difficulty following the distinctions the authors were making between these two metrics (niche overlap and niche tracking). Consequently, it is difficult to assess whether the difference between Figure 3 and Figure 4 is actually a biological difference, or simply a consequence of different methodologies that are expected to produce somewhat different results. This lack of clarity is due to the fact that the authors do not do an adequate job of connecting their methodology with their hypotheses. More explanation is needed. I did like the creative design of Figures 3 and 4, I just found the insights they were supposed to provide to be elusive.

Third, much of the results are focused on individual- versus population-level differences. However, as far as I can tell, the authors never explain what kind of individual variation they are measuring. I fail to understand whether the individual variation is across years, or just between seasons but within a year. And, if a bird was tracked for multiple years but the focus is on between-seasons, how was cross-year data treated in terms of non-independence?

Fourth, given that this paper focuses on one species, more biology of this species is needed in the paper. For example, Figure 2 shows some interesting dynamics going on in central Africa, which seem like they could have implications for tracking of climatic conditions, but this is never explained. Do White Storks have genetically encoded migration as juveniles, or do they follow their parents?

Finally, this comment represents a broader issue for the field that I think the authors should consider. The authors write on line 282-284: “The large within and between- individual variation found here indicate that population-level analyses could hide some flexibility at the individual level that could play an essential role in determining the resilience of species and populations to environmental change [48,49]”. OK...but it seems to me that every study that tracks individual migratory birds, which usually have sample sizes of 50 or less, identify variation in individual behavior and come to the conclusion that migration is more variable than typically assumed. But, variable compared to what? In a small sample, there will obviously be variation in how birds get to Africa and back. But the question is, what does the distribution of this behavior for the species as a whole look like? So if the point of this paper is to provide insight into this important question by identifying patterns of scale, I think it is not adequate to simply conclude that population-level analyses hide individual variation. In other words, I’m not clear on how this study accomplishes its important goals of using individual variation to reveal emergent patterns. Perhaps it’s just a matter of a better explanation.

Minor comments:

I was confused about how the two wintering populations were treated. In the introduction it suggests that these were lumped into one analysis, but in the results and Discussion (e.g., line 286) the authors start talking about the differences between the populations again as though this dimension is an advantage of the study. This was unclear.

Line 141-144. I don’t understand how the background space was created around the tracks. What is the space represented by “all the spatial and temporal extent that GPS locations included in this study”? A continent? Or less? Given the big debates in the niche modeling literature about selection of background points, this seems pretty important to explain.

Line 150-161: I just found this paragraph to be difficult to parse and the writing not very clear. It would be better if the authors would connect each method back to the hypotheses they are testing, which are never really stated.

Figure 1a: I found this confusing, because the middle panel’s illustration looks like low population overlap but the arrows label it as high population overlap. So I was never really clear on how the study identifies patterns of scale.

Author's Response to Decision Letter for (RSPB-2020-0658.R0)

See Appendix A.

RSPB-2020-1799.R0

Review form: Reviewer 1 (Camila Gómez)

Recommendation

Accept with minor revision (please list in comments)

Scientific importance: Is the manuscript an original and important contribution to its field?

Good

General interest: Is the paper of sufficient general interest?

Good

Quality of the paper: Is the overall quality of the paper suitable?

Good

Is the length of the paper justified?

Yes

Should the paper be seen by a specialist statistical reviewer?

No

Do you have any concerns about statistical analyses in this paper? If so, please specify them explicitly in your report.

No

It is a condition of publication that authors make their supporting data, code and materials available - either as supplementary material or hosted in an external repository. Please rate, if applicable, the supporting data on the following criteria.

Is it accessible?

Yes

Is it clear?

Yes

Is it adequate?

Yes

Do you have any ethical concerns with this paper?

No

Comments to the Author

The authors have done a very good job in addressing all the previous reviewer comments and I think the manuscript has improved substantially as a result. I was very happy to see the improvement in the figures and discussion.

I only spotted a few typos, so I urge authors to carefully revise throughout to make sure these and possibly others are corrected.

Lines 252-53: 'atfor' - should be 'at', and there is an extra 'individuals' in that sentence.

Line 280: 'andclimate' - separate

Review form: Reviewer 2

Recommendation

Accept with minor revision (please list in comments)

Scientific importance: Is the manuscript an original and important contribution to its field?

Excellent

General interest: Is the paper of sufficient general interest?

Excellent

Quality of the paper: Is the overall quality of the paper suitable?

Excellent

Is the length of the paper justified?

Yes

Should the paper be seen by a specialist statistical reviewer?

No

Do you have any concerns about statistical analyses in this paper? If so, please specify them explicitly in your report.

No

It is a condition of publication that authors make their supporting data, code and materials available - either as supplementary material or hosted in an external repository. Please rate, if applicable, the supporting data on the following criteria.

Is it accessible?

Yes

Is it clear?

Yes

Is it adequate?

Yes

Do you have any ethical concerns with this paper?

No

Comments to the Author

White Stork revision

I reviewed the original submission and provided detailed feedback. I find that the authors have done a superb job with this revision and that it is greatly improved. The methods are much clearer and thorough, and consequently the results are more interesting and accessible. The new figures are much more useful. The paper will make a valuable contribution to the literature as it is probably the first study to test for scaling in the tracking of environmental conditions during migration from individuals to populations. I have a few minor comments and questions:

- 1) I still find the illustration of H2 in Fig. 1 to be somewhat confusing, because the solid black curve for summer is drawn well outside the colored curves for summer, and likewise for winter. I have a hard time understanding what is proposed here. Perhaps describing a hypothetical example of how this could operate in practice (in a bird population) would be helpful.
- 2) Line 198-199: For the simulations, were the densities of occurrence points also divided by the densities of background points, as they were for the GPS data (line 180-182)? Or is this not necessary for some reason? Another line of explanation/clarification would be helpful here.
- 3) Line 250: I find it surprising that populations have higher niche overlap for weather than they do for climate. It would be helpful if the authors could provide some interpretation for this result. Is it biological? Or potentially an artifact (see comment 4, below)?

Fig 3: This figure helps a lot to explain the results. Good idea.

- 4) The higher tracking of weather and lower tracking of climate in western migrants compared to eastern migrants was attributed to anthropogenic affects. However, could this not also be due simply to autocorrelative effects from a marked difference in migratory distance between eastern and western populations? The western populations are short-distance migrants and thus may be

expected to follow local weather cues during migration than the long-distance eastern migrants which may cue more into photoperiod, etc. Additionally, the Iberian peninsula in the northern winter may be more dissimilar in climate to the breeding grounds in summer than is southern Africa, a pattern that could emerge at the scale of climatic niche. Do any of the analyses shed light on this possibility? I think this point should be considered as it is potentially very important when scaling these analyses up to multiple populations and species, as the others recommend in the Discussion.

Finally, the manuscript should be read carefully for typos, as I found several, especially in the revised parts.

Decision letter (RSPB-2020-1799.R0)

13-Aug-2020

Dear Dr Fandos

I am pleased to inform you that your manuscript RSPB-2020-1799 entitled "Seasonal niche tracking of climate emerges at the population level in a migratory bird" has been accepted for publication in Proceedings B.

The referee(s) have recommended publication, but also suggest some minor revisions to your manuscript. Therefore, I invite you to respond to the referee(s)' comments and revise your manuscript. Because the schedule for publication is very tight, it is a condition of publication that you submit the revised version of your manuscript within 7 days. If you do not think you will be able to meet this date please let us know.

- 1) A text file of the manuscript (doc, txt, rtf or tex), including the references, tables (including captions) and figure captions. Please remove any tracked changes from the text before submission. PDF files are not an accepted format for the "Main Document".
- 2) A separate electronic file of each figure (tiff, EPS or print-quality PDF preferred). The format should be produced directly from original creation package, or original software format. PowerPoint files are not accepted.
- 3) Electronic supplementary material: this should be contained in a separate file and where possible, all ESM should be combined into a single file. All supplementary materials accompanying an accepted article will be treated as in their final form. They will be published

alongside the paper on the journal website and posted on the online figshare repository. Files on figshare will be made available approximately one week before the accompanying article so that the supplementary material can be attributed a unique DOI.

If you wish to submit your data to Dryad (<http://datadryad.org/>) and have not already done so you can submit your data via this link [http://datadryad.org/submit?journalID=RSPB&manu=\(Document not available\)](http://datadryad.org/submit?journalID=RSPB&manu=(Document not available)) which will take you to your unique entry in the Dryad repository. If you have already submitted your data to dryad you can make any necessary revisions to your dataset by following the above link. Please see <https://royalsociety.org/journals/ethics-policies/data-sharing-mining/> for more details.

Sincerely,

Dr Locke Rowe

Reviewer(s)' Comments to Author:

Referee: 1

Comments to the Author(s).

The authors have done a very good job in addressing all the previous reviewer comments and I think the manuscript has improved substantially as a result. I was very happy to see the improvement in the figures and discussion.

I only spotted a few typos, so I urge authors to carefully revise throughout to make sure these and possibly others are corrected.

Lines 252-53: 'atfor' - should be 'at', and there is an extra 'individuals' in that sentence.

Line 280: 'andclimate' - separate

Referee: 2

Comments to the Author(s).

White Stork revision

I reviewed the original submission and provided detailed feedback. I find that the authors have done a superb job with this revision and that it is greatly improved. The methods are much clearer and thorough, and consequently the results are more interesting and accessible. The new figures are much more useful. The paper will make a valuable contribution to the literature as it is probably the first study to test for scaling in the tracking of environmental conditions during migration from individuals to populations. I have a few minor comments and questions:

1) I still find the illustration of H2 in Fig. 1 to be somewhat confusing, because the solid black curve for summer is drawn well outside the colored curves for summer, and likewise for winter. I have a hard time understanding what is proposed here. Perhaps describing a hypothetical example of how this could operate in practice (in a bird population) would be helpful.

2) Line 198-199: For the simulations, were the densities of occurrence points also divided by the densities of background points, as they were for the GPS data (line 180-182)? Or is this not necessary for some reason? Another line of explanation/clarification would be helpful here.

3) Line 250: I find it surprising that populations have higher niche overlap for weather than they do for climate. It would be helpful if the authors could provide some interpretation for this result. Is it biological? Or potentially an artifact (see comment 4, below)?

Fig 3: This figure helps a lot to explain the results. Good idea.

4) The higher tracking of weather and lower tracking of climate in western migrants compared to eastern migrants was attributed to anthropogenic affects. However, could this not also be due simply to autocorrelative effects from a marked difference in migratory distance between eastern and western populations? The western populations are short-distance migrants and thus may be expected to follow local weather cues during migration than the long-distance eastern migrants which may cue more into photoperiod, etc. Additionally, the Iberian peninsula in the northern winter may be more dissimilar in climate to the breeding grounds in summer than is southern Africa, a pattern that could emerge at the scale of climatic niche. Do any of the analyses shed light on this possibility? I think this point should be considered as it is potentially very important when scaling these analyses up to multiple populations and species, as the others recommend in the Discussion.

Finally, the manuscript should be read carefully for typos, as I found several, especially in the revised parts.

Author's Response to Decision Letter for (RSPB-2020-1799.R0)

See Appendix B.

Decision letter (RSPB-2020-1799.R1)

01-Sep-2020

Dear Dr Fandos

I am pleased to inform you that your manuscript entitled "Seasonal niche tracking of climate emerges at the population level in a migratory bird" has been accepted for publication in Proceedings B.

Your article has been estimated as being 8 pages long. Our Production Office will be able to confirm the exact length at proof stage.

Open Access

Paper charges

Sincerely,
Editor, Proceedings B
<mailto:proceedingsb@royalsociety.org>

Appendix A

Response to Editor and Reviewers on Manuscript ID RSPB-2020-0658

Comments by Editor:

This is an interesting study seeking to examine whether patterns of climatic niche tracking across seasons often seen in migratory animals reflect tracking of conditions by individuals or rather reflect an emergent property of populations within which individuals may show varying strategies. Solving this question would appear to be of broad relevance to understanding issues in the ecology, evolution, and conservation of migrants. Despite the appealing overall question (and the impressive tracking data set for White Storks), and that reviewers found considerable merit in the study, the manuscript as presented falls short in terms of conveying a clear message for various reasons. As noted by the reviewers, figures are unclear or seemingly not directly relevant to making the main points; they might be, but several improvements are necessary to make them understandable and compelling. The reviewers focused on figures showing results, but I also felt that the conceptual figure 1 could be substantially improved not only in terms of graphics but with a more informative legend so that alternative scenarios could be readily understood. In addition, as noted by one of the reviewers, important clarifications are required so that the reader can properly assess the conclusions and the impact of this work. Specific points include, but are not limited to, the distinctions between weather and climate and niche overlap and niche tracking, and clarity about the kind of individual variability being measured. To the fine comments provided by reviewers, I would add that framing the introduction (and the rest of the paper) around clearly articulated hypotheses with their specific predictions would likely help considerably to present a more coherent story.

Response

Thank you for the constructive reviews of our paper, as well as for the opportunity to resubmit the manuscript. We took great care to incorporate the valuable comments by the reviewers and editor and have considerably revised our methods, and improved the method descriptions as well as added new visualisations of the results to improve the understanding of our manuscript.

First, we reran the entire analyses with two major changes. Following Reviewer 2 comments, we divide the population between Western and Eastern migrants, depending on the migratory flyway that they used. In consequence, we used different background and study area for both populations in the niche overlap and niche tracking analysis. Although not requested by the reviewers, we now leave one-month gaps between seasons and thus more conservatively define the breeding, migration and wintering periods. Our results were robust to these changes, and we thus only show the results of these new data preparations and analyses. In the map of Figure 2, we now distinguish the annual cycle stages as suggested by Reviewer 1. The revised results continue highlighting the emergent pattern of climatic niche tracking at the population level.

Second, to address the Editor's and Reviewer's suggestion, we modified and improved all figures. In particular, we corrected the Fig.1 and improved the legend. In the new Fig. 2, we simplified the plot. Niche overlap is now shown in boxplots overlaid with jittering points

presenting the variation in niche overlap values as suggested by Reviewer 1. Additionally, we simplified Figure 3 to highlight the proportion of niche tracking at the different ecological and environmental scales.

Third, we thoroughly revised the Introduction and Discussion section in light of the Editors' and Reviewer 2's helpful comments, ensuring that the concepts are better presented, and the conclusions can be assessed according to our analyses and results.

Finally, we have revised our statistical analyses to test the individual and population variation and their potential role in the emergent patterns. Now, we used a mixed-model framework to explore the importance of within-individual and between-individual as well as within-population and between-population variation in niche overlap and niche tracking. By doing that, now, we are able to evaluate how white storks divide the phenotypic expression of these traits among different hierarchical levels, which could have different evolutionary consequences.

We very much appreciate the thorough and thoughtful comments, which helped us to improve our manuscript. Below, we address each comment in more detail.

Comments by Reviewer 1:

Comments to the Author(s)

This manuscript describes how niche tracking varies across spatio-temporal scales and between individuals and populations of White Storks. The detailed multi-year tracking dataset that they used is very impressive and allows for unprecedented detail to be included in these types of analysis. I thought the paper was very interesting and has great potential to be a valuable contribution in its field.

Given this amazing dataset, I was a bit disappointed by the main figures on the ms, and think that the authors should explore other options to improve the visualisation their results. In my opinion, improving those figures will make the paper much more compelling and its main message clearer to a wider readership. I think it is critical that authors show the variability in their dataset explicitly, hence not just graphing means, but also raw data points, and provide a clearer framework for the interpretation of their figures.

Thank you for your very thorough and constructive review, valuable feedback and time spent on supporting us to improve our manuscript.

I have provided some details below and hope they can serve to improve the manuscript.

Notes:

Title – I would change 'migratory birds' to 'a migratory bird'. This is a single species study.

Good point. We edited this accordingly.

Figure 2 (and figure SM4). Very cool but completely unnecessary in the context of your manuscript. They do not add anything to your message. Maybe a way to include a map of tracks could be if you color code the tracks by season. That map could then be included as a small side panel within a main figure of niche overlap, just showing where geographically are the storks during each period in their annual cycle. You could add a small illustration of a White Stork to make it even better!

We added a small panel to Figure 3 showing where white storks were geographically located during each period in their annual cycle.

Figs 3 and 4. These are the most important figures but I had a lot of trouble interpreting them. I like the fact that they are circular, but I don't like that they don't make the variability of the data explicit, and frankly, I did not understand the idea with the line colors and how they connect to each other. I think that authors could keep these (maybe as a side or complementary panel to the figure I am suggesting), and providing a clearer explanation of their interpretation. I strongly suggest authors explore other attractive ways to visualise their data. For instance, they could explore creating a kernel density circular plot for individuals and populations, showing all the individual data points and estimating the core D scores from the seasonal distributions. Points would be color and spatially coded on the circles by season of the year (date would determine the cartesian angle of each point) and D value would determine the distance from the center of the circle. I imagine four 'bubbles', one for each season, and with points concentrating around the mean D score for the season, but with the variability explicitly shown by all the data points. Showing a spatial representation of this variability is a much more transparent and thorough way to present these data, and I think will be better for interpretation.

Thank you for pointing out this unclarity and for the great suggestion. We modified and simplified both figures to explicitly show the variability and to improve the essential interpretation of both.

Fig 5. Please overlay all the jittered data points on the boxplots.

We changed this accordingly.

Line 260 – 'tracking its climatic niche' should be 'tracking their climatic niche'

We edited this accordingly.

Line 272 – 284. The point raised in this paragraph should be emphasised more as it is one of the most novel aspects of this study. There has rarely been data available to look and intra individual and inter individual variation in migration strategies, so they should make a strong point about that here.

Thank you for your suggestion. In the revised manuscript, we used a different approach to explore the individual and population-level variation. We now used a mixed-model framework to analyse the within and between-individual and the within and between-population variation, respectively. Additionally, we modified the Discussion and now more strongly emphasise the potential eco-evolutionary consequences of the individual plasticity in seasonal niche overlap and niche tracking.

Line 299 – Individuals should be 'individual'

We edited this accordingly.

Comments by Reviewer 2:

This paper uses detailed tracking data to assess the variability of White Stork migration and

whether individuals, as well as the population as a whole, tracks climatic conditions throughout the annual cycle. The objective of the paper is valuable and the potential for interesting insights is strong. Overall, the presentation of the results and the writing was pretty good. However, I felt that the paper lacked important detail and explanation in several key areas that make it very difficult for the reader to assess the conclusions and the impact of this work. My comments are intended to point out areas that need greater explanation or thought.

Thank you for raising a number of valuable points here. We address each of these below:

Major comments:

First, the authors assess the impact of weather and climate, but they don't adequately explain the biological difference between the two or how their data allow them to make this distinction for the birds.

We apologise for not being sufficiently clear here. In the revised manuscript, we provided a clearer explanation of the differences between climate and weather from a biological perspective. We derived the climate data as a long-term average over > 15 years, and weather data as the fine-scale conditions over a short period; < 20 days.

White stork migration and movement are characterised by processes that act over multiple spatial and temporal scales. By considering weather and climate, we want to be sure that we use the correct environmental scale that is relevant to the white stork's movement decisions and in consequence, the seasonal niche patterns. Individual movement is influenced by local cues (food resources) strongly correlated with the weather conditions. At the same time, individuals are influenced by the spatial distribution of populations and the connectivity of the seasonal areas that are constrained by climatic conditions. We revised our manuscript to make a more explicit link between our hypothesis of the seasonal niche patterns and environmental spatiotemporal scale (weather/climate).

Second, much of the paper is focused on contrasting niche overlap with niche tracking. However, the distinction between these two concepts is not explained. Even as someone who is generally familiar with niche modelling and has read most of the papers on niche tracking in migratory birds, I had difficulty following the distinctions the authors were making between these two metrics (niche overlap and niche tracking). Consequently, it is difficult to assess whether the difference between Figure 3 and Figure 4 is actually a biological difference, or simply a consequence of different methodologies that are expected to produce somewhat different results. This lack of clarity is due to the fact that the authors do not do an adequate job of connecting their methodology with their hypotheses. More explanation is needed. I did like the creative design of Figures 3 and 4, I just found the insights they were supposed to provide to be elusive.

We now improved the description of these methods. We first calculated niche overlap between seasons and, second, we estimated niche tracking by similarity tests that evaluate whether niche overlap was higher than expected by chance by means of null models. We tried to make this distinction clear in the text now.

Although the hypothesis for niche overlap and niche tracking is the same, we have different biological interpretations for both. Specifically, individuals or populations may show low seasonal niche overlap if the available environmental conditions between seasons differs

strongly. Here, the similarity analysis will show us if there is an apparent selection for a different or similar environment taking into account the availability of environment.

Third, much of the results are focused on individual- versus population-level differences. However, as far as I can tell, the authors never explain what kind of individual variation they are measuring. I fail to understand whether the individual variation is across years, or just between seasons but within a year. And, if a bird was tracked for multiple years but the focus is on between-seasons, how was cross-year data treated in terms of non-independence?

Thank you for pointing this out. Now, we used a mixed-model framework to explore the importance of within-individual and between-individual variation as well as within-population and between-population variation in total niche overlap and niche tracking. We develop different models for populations and individuals, and for each environmental scale using both niche overlap (continuous; between 0-1) and niche tracking (binomial; tracking or not) as response variables. This is now detailed in the Method's section.

We estimate the individual contribution by including the ID of each individual as random effect. This approach allowed us to model differences in mean response between individuals. A residual error is also estimated, representing the within-individual variance. For the population-level analysis, we included population ID as a random effect (Western or Eastern). This model allowed us to model the differences between populations, and in this case, the residual error is the within-population variance.

Following your suggestion, we included the nested effect of the year in both population and individual-level analyses to control for the potential non-independence between cross-year data.

With this new analysis, we are now able to disentangle the different components of variation in seasonal niche overlap and niche tracking, and, therefore, discuss the potential roles of this plasticity to develop emergent patterns.

Fourth, given that this paper focuses on one species, more biology of this species is needed in the paper. For example, Figure 2 shows some interesting dynamics going on in central Africa, which seem like they could have implications for tracking of climatic conditions, but this is never explained. Do White Storks have genetically encoded migration as juveniles, or do they follow their parents?

Thank you for your suggestion. In light of this comment, we added in the Introduction some more information on the biology of the White stork migration, helping to set the context of our hypothesis.

White Storks are frequently used as study models as detailed knowledge on their migratory behaviour and migratory patterns is available. However, it is unclear to what extent all these migratory patterns in White storks are controlled by inherited genetic information (sensu Mueller et al. 2011). White storks are obligatory social migrants where juvenile and adults migrate together in mixed flocks during the autumn migration. In that sense, juveniles have higher energy expenditure compared to adults. Therefore, for this study, we only included adult white storks to avoid patterns related to poor flight skills of juvenile storks.

Finally, this comment represents a broader issue for the field that I think the authors should consider. The authors write on line 282-284: "The large within and between- individual variation found here indicate that population-level analyses could hide some flexibility at the individual level that could play an essential role in determining the resilience of species and

populations to environmental change [48,49]". OK...but it seems to me that every study that tracks individual migratory birds, which usually have sample sizes of 50 or less, identify variation in individual behavior and come to the conclusion that migration is more variable than typically assumed. But, variable compared to what? In a small sample, there will obviously be variation in how birds get to Africa and back. But the question is, what does the distribution of this behavior for the species as a whole look like? So if the point of this paper is to provide insight into this important question by identifying patterns of scale, I think it is not adequate to simply conclude that population-level analyses hide individual variation. In other words, I'm not clear on how this study accomplishes its important goals of using individual variation to reveal emergent patterns. Perhaps it's just a matter of a better explanation.

We realise that this statement was formulated too generally. To remedy this, we have reanalysed (see above) and expanded the Discussion to explain better the goal of analysing the variation in seasonal niche overlap and tracking at both individual and population level. Although there is evidence that white storks have significant breeding site philopatry and follow remarkably similar migration routes over many years, these studies usually focus on the individual migratory variation on the timing and geographic patterns of their migrations. However, to our knowledge, our study is the first to explore the seasonal niche tracking patterns at multiple ecological levels,

We explored the individual and population-level variation on niche overlap, to disentangle the different component of these traits variation across ecological levels and environmental scale (weather and climate). If climatic niche tracking at the population level is an emergent pattern, this should be a result of individual variation. The amount of observed individual plasticity in seasonal niche tracking can result from (1) differences among individuals, and (2) differences within individuals, which are expected to influence the individual's ability to respond to environmental changes. In conclusion, species or population-level approaches will only provide a partial picture of the seasonal niche tracking patterns if variation in individual-level niche overlap and niche tracking is high.

Minor comments:

I was confused about how the two wintering populations were treated. In the introduction it suggests that these were lumped into one analysis, but in the results and discussion (e.g., line 286) the authors start talking about the differences between the populations again as though this dimension is an advantage of the study. This was unclear.

Thank you for your comment. We rerun the analysis separating Western and Eastern migratory populations by creating a different background for each population. The individual's results are consistent with our previous conclusions, and with this new analysis, we are able to add another layer of complexity calculating and comparing populations with different migratory flyways. In particular, we show that the Western migrants show less climatic niche tracking than their Eastern counterparts. We interpret this different degree of niche tracking as linked to recent anthropogenic changes and constant spatiotemporal availability of food resources (e.g. from landfills).

Line 141-144. I don't understand how the background space was created around the tracks. What is the space represented by "all the spatial and temporal extent that GPS locations included in this study"? A continent? Or less? Given the big debates in the niche modeling

literature about selection of background points, this seems pretty important to explain.

We agree that it is essential to point out the background selection criteria that we follow during our study. In this new version, we created different backgrounds depending on the migratory route used by the individuals, east or west. As our goal was not to generate a proper distribution model, and we are not going to extrapolate beyond sample units, we generated the background locations following inclusive criteria. We placed points randomly within the Maximum Convex Polygon (MCP) around the GPS locations plus a buffer of 300 km (maximum home range of White Storks; Zurell et al. 2018). Furthermore, as we are considering the temporal dimension, all background points were attached to a seasonal specific random date in the same temporal range of the GPS locations. To explain this better, we revised the Methods section.

Line 150-161: I just found this paragraph to be difficult to parse and the writing not very clear. It would be better if the authors would connect each method back to the hypotheses they are testing, which are never really stated.

Sorry for not being sufficiently clear here. In the revised manuscript, we make a more explicit connection between each Method and hypothesis.

Figure 1a: I found this confusing, because the middle panel's illustration looks like low population overlap but the arrows label it as high population overlap. So I was never really clear on how the study identifies patterns of scale.

Good point. We modify the middle panel's illustration in Fig. 1a to show clearly a high population overlap.

Appendix B

Response to Editor and Reviewers

Thank you for all the valuable comments. We are pleased to see that our manuscript has been accepted for publication in Proceedings B.

We carefully revise the manuscript to remove the typos. Below, we address each comment of Reviewer 2 in more detail.

Comments by Reviewer 2:

- 1) I still find the illustration of H2 in Fig. 1 to be somewhat confusing, because the solid black curve for summer is drawn well outside the coloured curves for summer, and likewise for winter. I have a hard time understanding what is proposed here. Perhaps describing a hypothetical example of how this could operate in practice (in a bird population) would be helpful.

Thank you for pointing out this unclarity. We revised Figure 1, adding a new “individual” curve such that the solid black curve (and the dashed black curve, respectively) now describe the minimum and maximum niche values found across all individual (coloured) curves.

- 2) Line 198-199: For the simulations, were the densities of occurrence points also divided by the densities of background points, as they were for the GPS data (line 180-182)? Or is this not necessary for some reason? Another line of explanation/clarification would be helpful here.

Thank you for pointing this out. The reviewer is absolutely right that the occurrence densities were also corrected by the background densities. We added a clarification where we highlight that the occurrence points are also divided by the densities of background points in the simulations.

- 3) Line 250: I find it surprising that populations have higher niche overlap for weather than they do for climate. It would be helpful if the authors could provide some interpretation for this result. Is it biological? Or potentially an artifact (see comment 4, below)?

Thanks again for this comment. We address it in more detail below in response to the fourth point made by the reviewer.

Fig 3: This figure helps a lot to explain the results. Good idea.

Thanks!

- 4) The higher tracking of weather and lower tracking of climate in western migrants compared to eastern migrants was attributed to anthropogenic affects. However, could this not also be due simply to autocorrelative effects from a marked difference in migratory distance between eastern and western populations? The western populations are short-distance migrants and thus may be expected to follow local weather cues during migration than the long-distance eastern migrants which may cue more into photoperiod, etc. Additionally, the Iberian peninsula in the northern winter may be more dissimilar in climate to the breeding grounds in summer than is southern Africa, a pattern that could emerge at the scale of climatic niche. Do any of the analyses shed light on this possibility? I think this point should be considered as it is potentially very important when scaling these analyses up to multiple populations and species, as the others recommend in the Discussion.

Sorry for not being sufficiently clear here. As you mentioned there is different predictions on the relationship between environmental variation, movement and niche characteristics (Laube et al. 2015). Therefore, to control the potential effects of the migratory distance, we used the robust method to measure niche overlap and niche tracking developed by Broennimann et al. 2012. This framework accounts for biases introduced by geographical autocorrelation by assessing whether niche overlap is higher than expected by available environment, making it a suitable approach for analysing niche differences between species, subspecies or individuals that differ in their geographical distributions. Thus, the approach chosen controls for differences in migratory distance and geographical location and we attribute the resulting differences in climatic niche tracking between Western and Eastern migrants to anthropogenic effects. As the reviewer rightly points out, the underlying hypotheses for low or high niche tracking may differ between short-distant western vs. long-distant eastern migrants. Yet, it is well established that western migrants have evolved the shorter-distance migration more recently while historically wintering in Western Africa, or turned into residents in response to anthropogenic environmental change (Flack et al. 2016). These anthropogenic effects are thus a likely reason for lower niche tracking in western migrants.

Finally, the manuscript should be read carefully for typos, as I found several, especially in the revised parts.

We carefully revised the manuscript.